# Changes in Oral Microbial Diversity in a Piglet Model of Traumatic Brain Injury

**DOI:** 10.3390/brainsci12081111

**Published:** 2022-08-21

**Authors:** Julie Heejin Jeon, Jeferson M. Lourenco, Madison M. Fagan, Christina B. Welch, Sydney E. Sneed, Stephanie Dubrof, Kylee J. Duberstein, Todd R. Callaway, Franklin D. West, Hea Jin Park

**Affiliations:** 1Department of Nutritional Sciences, College of Family and Consumer Sciences, University of Georgia, Athens, GA 30602, USA; 2Department of Animal and Dairy Science, College of Agricultural and Environmental Sciences, University of Georgia, Athens, GA 30602, USA; 3Regenerative Bioscience Center, University of Georgia, Athens, GA 30602, USA

**Keywords:** oral microbiome, neurological disease, porcine model

## Abstract

Dynamic changes in the oral microbiome have gained attention due to their potential diagnostic role in neurological diseases such as Alzheimer’s disease and Parkinson’s disease. Traumatic brain injury (TBI) is a leading cause of death and disability in the United States, but no studies have examined the changes in oral microbiome during the acute stage of TBI using a clinically translational pig model. Crossbred piglets (4–5 weeks old, male) underwent either a controlled cortical impact (TBI, *n* = 6) or sham surgery (sham, *n* = 6). The oral microbiome parameters were quantified from the upper and lower gingiva, both buccal mucosa, and floor of the mouth pre-surgery and 1, 3, and 7 days post-surgery (PS) using the 16S rRNA gene. Faith’s phylogenetic diversity was significantly lower in the TBI piglets at 7 days PS compared to those of sham, and beta diversity at 1, 3, and 7 days PS was significantly different between TBI and sham piglets. However, no significant changes in the taxonomic composition of the oral microbiome were observed following TBI compared to sham. Further studies are needed to investigate the potential diagnostic role of the oral microbiome during the chronic stage of TBI with a larger number of subjects.

## 1. Introduction

The oral microbiome is the second most diverse in the human body after the gut and is usually stable and nonpathogenic in healthy individuals [1]. The oral cavity is a significant place for the microbial community and a door of invading pathogens and toxic substances from outside of the body [2]. Bacteria are the main inhabitants on the oral cavity and colonize on two types of surfaces: the soft (oral mucosal) and the hard tissue (teeth) [2]. Unlike the bacteria in the gut, which reside in the intestinal cavity with mucus, bacteria in the oral cavity form a bacterial biofilm and coat the oral cavity surfaces, mainly dominated by the genus *Streptococcus* [1,3]. The biofilms maintain a homeostatic balance with their host under normal condition; however, it becomes dysbiotic in disease status by altering the microbial composition or by loss of diversity [4]. These statuses can cause an imbalance in the oral microbiome, which is detrimental to oral health and linked to increased risk of systemic diseases [5].

Recently, researchers have shown the potential role of a diagnostic tool of oral microbes in neurological diseases, with studies highlighting dynamic changes in the oral microbiome of Alzheimer’s disease (AD) and Parkinson’s disease (PD) patients. Wu et al. [6] reported that AD patients had significant increases in Firmicutes, *Lactobacillus*, and *Streptococcus* levels in dental plaque compared to that in healthy controls. *Lactobacillus*, which is known as a probiotic in the gut microbiome, is considered pathogenic bacteria in the oral cavity since it causes root caries and tooth loss [7,8]. Similarly, *Streptococcus*, the most commonly found in the oral microbiome, also contains some pathogenic species such as *Streptococcus mutans*, which is an acid-producing bacteria that causes carious lesions [9]. Consistent with these observations, more missing teeth and higher dental plaque weight were found in AD patients compared to healthy controls, and poor oral condition was correlated with cognitive deficits [6], suggesting a potential link between the existence of periodontal pathogens and AD development. Similarly, patients with PD had increased oral microbial levels of *Streptococcus* (*Streptococcus mutans* [10] and *Streptococcus pneumonia* [11]), *Lactobacillaceae* [10,12], and *Lactobacillus* [11,13], and study has shown a positive correlation between *Lactobacillus reuteri* and slower movement in PD patients [13]. These studies suggest that dysbiosis of the oral microbial community may be closely related to cognitive and functional decline in neurological conditions and may serve as biomarkers of these conditions.

Traumatic brain injury (TBI) is one of the most common types of brain injuries and often leads to major disability and death in global populations [14]. Despite the devastating effects of TBI, there are a number of challenges, including detection of mild TBI, discerning TBI severity, and predicting potential patient outcomes. Understanding the dynamic interplay between TBI and the oral microbiome may provide unique insight into using the oral microbiome as a biomarker to detect TBI. However, the majority of published studies exploring the relationship between TBI and microbiome have almost exclusively focused on the lower intestine or fecal samples [15,16,17]. To the best of our knowledge, there are no studies that have explored the oral microbiome changes after TBI. Collecting fecal samples can be challenging, as patients with brain injury commonly experience bowel dysfunction, such as constipation [18]. Therefore, the highly accessible mucosal surface of the oral cavity is potentially a preferable microbiome for evaluating TBI patient injury.

In the current study, we used a well-established piglet model [19,20,21,22] to assess changes in oral microbial diversity and composition during the acute stage of TBI. The pig is a robust translational model due to its comparable brain and gastrointestinal (GI) anatomy, physiology, and pathophysiological traits relative to humans, thus making the pig an ideal large-animal model for clinical research [22,23]. The findings of this study may support the potential role of the oral microbiome as a diagnostic and prognostic biomarker for TBI and provide the basis for future oral microbiome studies in TBI patients.

## 2. Materials and Methods

### 2.1. Animals, TBI Induction, and Oral Mucosa Collection

Castrated, crossbred male piglets (*n* = 12; 4–5 weeks old) were acquired from the University of Georgia swine unit and randomly selected and assigned into two groups; (1) TBI surgery group (TBI, *n* = 6) or (2) sham surgery group (sham, *n* = 6). Moderate/severe TBI was induced using a controlled cortical impact (CCI) as previously described [21] with the following parameters velocity: 4 m/s, depth: 9 mm, dwell: 400 ms. The sham group underwent a craniectomy but did not receive a CCI. Induced TBI was confirmed by magnetic resonance imaging (MRI) 1 day post-surgery (data not shown). Oral mucosa was collected from the upper and lower gingiva, buccal mucosa, and floor of the mouth using sterile cotton swabs at pre-surgery and 1, 3, and 7 days post-surgery (PS). Samples were stored at −80 °C until further analysis. All work in this study was conducted in accordance with the guidelines established by the University of Georgia Institutional Animal Care and Use Committee.

### 2.2. Oral Microbial DNA Extraction and 16 s rRNA Gene Sequencing Analysis

Oral mucosa was dissolved into sterile PBS, and oral DNA extraction was processed according to a modified protocol of the QIAamp Fast DNA Stool Mini Kit (Qiagen; Germantown, MD, USA). The DNA concentration of each sample was quantified spectrophotometrically utilizing the Synergy H4 multimode plate reader (BioTek, Winooski, VT, USA). Then, 16s ribosomal RNA (rRNA) gene sequencing was conducted from the extracted DNA samples by LC sciences (Houston, TX, USA). The V3-V4 region was amplified with primer pairs S-D-Bact-0341-b-S-17 (5′-CCTACGGGNGGCWGCAG-3′) and S-D-Bact-0785-a-A-21 (5′-GACTACHVGGGTATCTAATCC-3′) and sequenced using the Illumina NovaSeq platform. Data were demultiplexed before being converted into FASTQ files, and the paired-end sequences were imported into QIIME 2 [24]. The non-biological nucleotides were then removed, and sequences were denoised, dereplicated, and chimera-filtered using DADA2 [25]. Taxonomies were assigned to the sequences by using a pre-trained naive Bayes classifier trained on the SILVA 138 SSU database [26], and reads were classified by taxon using the fitted classifier [27]. For alpha and beta diversity analyses, all samples were rarefied to a common sequencing depth. Alpha diversity was determined by the number of observed features (number of ASVs), Faith’s phylogenetic diversity (total length of phylogenetic branches), Shannon index (species richness and evenness), and Pielou’s evenness (species evenness). Beta diversity was assessed using the unweighted UniFrac distance matrix, which considers phylogenetic connections and was visualized by principal coordinate analysis (PCoA). Oral microbial composition was measured at phylum (relative abundance >1%), family (>1%), genus (>1%), and species (>0.5%) levels. All indices of microbial diversities and composition were examined using QIIME2 plugins.

### 2.3. Statistical Analysis

Mixed effects ANOVA was used to compare differences between groups and time points. The results showed the main effect of time (time effect) and group (group effect) and the interaction effect between time and group (time-by-group interaction effect). Data are shown as fitted mean ± standard error of the mean (SEM). Differences in beta diversity were evaluated by Bonferroni-corrected multiple comparisons (corrected *p*-value) between each time point and between the groups. *p*-values under 0.05 were regarded as significant for all statistical tests.

## 3. Results

### 3.1. Oral Microbial Diversity

The alpha diversity indexes were analyzed to examine changes in oral microbial diversity in TBI and sham piglets during the acute stage of TBI (Figure 1). Significant time effects were found in number of observed features (time effect *p* = 0.00), Shannon (time effect *p* = 0.008), evenness (time effect *p* = 0.043), and Faith’s phylogenetic diversity (time effect *p* = 0.00) up to 7 days post-surgery. However, there were no group or time-by-group interaction effects in alpha diversity indexes except for Faith’s phylogenetic diversity. Significant group (*p* = 0.003) and time-by-group interaction effects (*p* = 0.001) were observed in Faith’s phylogenetic diversity, showing significantly lower levels in the TBI group compared to the sham group at 7 days PS (Figure 1D). These results suggest that species diversity and evenness were not significantly altered after TBI, but the phylogenetic diversity differed 7 days post-surgery compared to the sham group.

Beta diversity of the oral microbiome was assessed using an unweighted UniFrac distance matrix to examine the similarity or dissimilarity of microbial patterns between the TBI and sham groups (Figure 2). Interestingly, both the TBI and sham groups had significantly different beta diversity between time points (Figure 2A), showing altered beta diversity at 1, 3, and, 7 days post-surgery compared to pre-surgery. The sham group also showed different beta diversity at 7 days post-surgery compared to 1 and 3 days post-surgery. These results indicate that the oral microbial structure was altered post-surgery in both the TBI and sham groups. Between the TBI and sham group, distinct microbial patterns were observed at 1, 3, and 7 days post-surgery, with the most apparent difference at 7 days post-surgery (Figure 2B, all Bonferroni corrected *p*-value < 0.05). This finding implies that TBI surgery altered oral microbial structure aside from the effects of surgical stress and that the changes persisted up to 7 days post-surgery.

### 3.2. Taxonomic Composition of the Oral Microbiome

Oral microbial composition (Figure 3) was measured at the phylum (*p*, relative abundance >1%), family (F, relative abundance >1%), genus (G, relative abundance >1%), and species level (S, relative abundance > 0.5%) in both TBI and sham groups pre-surgery and 1, 3, and 7 days PS. The oral microflora in this study showed a similar predominant bacterial composition compared to previous human and porcine oral microbiome studies [28,29,30]. The top five most abundant phyla pre-surgery were Firmicutes (Mean relative abundance ± SEM, 50.56 ± 4.08%), Proteobacteria (34.10 ± 4.09%), Actinobacteria (8.66 ± 1.72%), Bacteroidetes (4.6 ± 0.78%), and Fusobacteria (1.67 ± 0.20%). The top five most prevalent bacterial families pre-surgery were *Streptococcaceae* (27.03 ± 3.11%), *Pasteurellaceae* (16.39 ± 1.98%), *Moraxellaceae* (10.66 ± 1.96%), *Lactobacillaceae* (6.45 ± 1.91%), and *Veillonellaceae* (5.58 ± 1.12%). The top five most abundant bacterial genera pre-surgery were *Streptococcus* (27.03 ± 3.11%), *Actinobacillus* (14.68 ± 1.77%), *Moraxella* (6.95 ± 1.39%), *Lactobacillus* (6.44 ± 1.91%), and *Veillonella* (4.97 ± 1.04%). Lastly, the top five most prevalent bacterial species pre-surgery were *Streptococcus suis* (6.42 ± 0.95%), *Actinobacillus indolicus* (3.59 ± 0.74%), *Actinomyces denticolens* (3.49 ± 1.60%), *Actinobacillus porcitonsillarum* (3.08 ± 0.70%), and *Actinomyces howellii* (1.11 ± 0.49%).

Significant time effects were observed in several microbial taxonomic compositions including P_Firmicutes, Proteobacteria, Bacteroidetes, Fusobacteria, F_*Streptococcaceae*, *Pasteurellaceae*, *Lactobacillaceae*, *Lachnospiraceae*, *Micrococcaceae*, *Ruminococcaceae*, *Leptotrichiaceae*, G_*Streptococcus*, *Actinobacillus*, *Moraxella*, *Lactobacillus*, *Rothia*, *Blautia*, *Leptotrichia*, S_*Streptococcus suis*, *Actinobacillus indolicus*, and *Bergeyella porcorum* in both TBI and sham groups (data not shown, all *p* < 0.05). Among oral bacteria, significant time-by-group interaction effects were found in P_Proteobacteria (*p* = 0.026), F_*Leptotrichiaceae* (*p* = 0.037), F_*Micrococcaceae* (*p* = 0.019), and G_*Rothia* (*p* = 0.023) in both TBI and sham groups; however, the post hoc Tukey HSD (honestly significant difference) comparison did not show any differences between TBI and sham groups at any time point post-surgery. Therefore, the oral microbial composition did not differ between TBI and sham groups at the phylum (1>%), family (1>%), genus (1>%), or species (0.5>%) level during the acute stage of TBI, despite showing some differences in phylogenetic diversity and microbial patterns by alpha and beta diversity, respectively.

## 4. Discussion

Here, we investigated for the first time changes in the oral microbiome during the acute stage of TBI in a piglet model. Moderate/severe TBI induced differences in Faith’s phylogenic diversity and beta diversity between the TBI and sham piglets. The Faith’s phylogenetic diversity was lower in the TBI group at 7 days post-surgery compared to the sham group, and different beta diversity (microbial patterns) was found between the groups post-surgery, with the most distinct alterations at 7 days post-surgery. However, the taxonomic composition of the oral microbiome did not significantly change following TBI. This translational study provides novel insight into the oral microbiome TBI response and suggests that further studies are warranted to investigate changes in the oral microbiome during the chronic stage of TBI.

Dysbiosis, an indicator of an imbalanced microbiome, is characterized by a reduction in microbial diversity and compositional changes and is associated with various disease states [31]. Previous studies showed that AD patients had decreased microbial richness and diversity, with lower levels of oral microbial numbers of operational taxonomic units (OTUs) [6], Chao1 (species richness), Shannon index, and phylogenetic diversity whole tree as compared to healthy controls patients [32]. Similarly, PD patients had lower alpha diversity with reduced species richness and evenness as well as different microbial communities (beta diversity) compared to the healthy controls [10]. In this study, we examined the oral microbiota profile in a translational piglet TBI model and found Faith’s phylogenetic diversity only was lower in the TBI group 7 days post-surgery compared to the sham group, while number of observed features, Shannon, and evenness only had significant time effects. This suggests that TBI may have a limited effect on oral microbial diversity during the acute stage of TBI. Faith’s phylogenetic diversity represents species richness, computing the sum of the branches of phylogenetic trees connecting all species of a given taxonomic group [33]. Faith’s phylogenetic diversity is known to be more sensitive in distinguishing disease factors in humans, as this diversity measures phylogenetic differences between species, whereas traditional species diversity does not differentiate between species [34,35]. Previous studies showed that phylogenetic diversity was reduced in the oral microbiome of patients with AD [32] and nasopharyngeal carcinoma [36] compared to the healthy controls, which was also shown in the gut microbiome of patients with AD [37], inflammatory bowel disease [38], and autism [39]. These results suggest that decreased phylogenetic diversity may be linked to poor microbial resilience and health. Therefore, reduced phylogenetic diversity can be a potential indicator of TBI, but further studies are needed to clarify its role during the acute stage of TBI.

Beta diversity represents similarity or dissimilarity in microbial structure between groups that captures the changes in the microbial composition [40]. The current study showed that the most distinct differences were found 7 days post-surgery between the TBI and sham groups, similar to Faith’s phylogenetic diversity. Both UniFrac matrix of the beta diversity and Faith’s phylogenetic diversity measure phylogenetic distances between and within the communities, respectively; therefore, differential microbial patterns showed by the beta diversity at 7 days post-surgery likely have a relationship with the changes in phylogenetic diversity between the groups. Previous studies showed that patients with PD [10,11,12] or depression [41] had an alteration in oral bacterial ecology (beta diversity) compared to healthy controls. Interestingly, those PD patients did not have more decayed teeth or periodontitis [10], or their plaque index scores improved to a comparable level to that of healthy controls after through toothbrushing [11], implying that the altered beta diversity may be caused by the disease itself rather than inadequate dental hygiene. PD patients also had high proportion of dental-caries-associated bacteria (*Streptococcus mutans* [10], *Lactobacillaceae* [12], and *Lactobacillus* [11]), periodontitis-associated bacteria (*Kingella oralis* [10], *Negativicutes*, and *Tannerella forsythia* [11]), and potential pathogenic bacteria (*Prevotella* and *Veillonella* [12]). These bacteria can cause chronic inflammation, which impairs the structure and function of oral tissues, leading to dental caries and periodontal disease [42,43]. The inflammatory response is proposed as a key factor in mediating the effects of oral pathogens on brain pathophysiology [44]. Therefore, the altered beta diversity after TBI shown in this study may suggest an increased risk of worsening oral conditions and an elevation in opportunistic oral pathogens as TBI progresses that will affect disease severity or recovery.

Although changes were observed in oral microbial diversity following TBI, the present study did not find significant differences in taxonomic composition between TBI and sham animals. There are a number of potential reasons as to why oral microbiota composition changes were not observed between TBI and the sham groups in the current study. First, dysbiosis induced by TBI may cause higher inter-individual variation in microbial composition than sham piglets. According to the Anna Karenina principle, the microbial community varies more in dysbiotic individuals than that of healthy individuals [45]. High variability may reduce the ability of bacteria to regulate microbial composition, preventing the establishment of unique microbial communities [45]. This theory was also applied in a study of relapse/refractory multiple myeloma patients who did not have a distinct oral microbial composition compared to the general population [46]. To overcome this challenge, it may be necessary to increase the number of animals to enhance the statistical power. Second, surgical stress may have had a major influence on microbial changes in both TBI and sham groups. It is expected that the significant time effects in the oral microbial composition may have been induced by surgical or environmental stress, as both can alter microbial diversity and microbial composition [47]. Existing studies reported that the Fusobacteria level was increased in response to treatment of stress hormone and cortisol in subgingival dental plaque samples cultured in vitro [48], and its genus level of oral *Fusobacterium* and *Leptotrichia* spp. were positively associated with cortisol levels in humans [49]. The present study also showed increased Fusobacteria post-surgery and *Leptotrichia* at 3 days PS in both TBI and sham groups compared to pre-surgery (data not shown). This suggests a potential impact of surgical stress on microbial changes in both the TBI and sham groups that may mask the compositional differences between groups. The precise mechanisms by which cortisol alters the oral microbiome are still unknown, but several stress hormones have been related to both stimulatory and inhibitory effects on oral microbial growth [50,51]. Future studies should consider this complexity of stress effect on oral microbiome changes. Third, significant changes in the oral microbiome may not be apparent at the acute TBI stage. Neurodegenerative diseases such as AD and PD are chronic diseases that develop slowly over time and affect primarily elderly people [52,53]. Studies of chronic AD and PD patients showed individuals possessed altered oral microbial diversity or abundance compared to healthy controls [6,10,11,12,13,32]. Similarly, diseases that have been reported to have altered oral microbiome are chronic inflammatory disorders (e.g., rheumatoid arthritis) or chronic diseases such as cancer, diabetes, and cardiovascular disease [54]. Therefore, the acute stage of TBI may have been too early to detect robust quantitative changes in alpha diversity and differences in phylum, family, genus, and species. It will be intriguing to investigate changes in the oral microbiota during the chronic stage of TBI. Finally, the changes in the oral microbiome composition may differ based on several factors, such as sampling location and the age of the pig. For example, in PD patients, changes in oral microbial diversity and composition were different between samples of soft (tongue dorsum and buccal mucosa) and hard tissue (chewing surfaces of the molars) of the oral cavity [11]. Moreover, the oral microbiota in infant and toddlers was less diverse than that of adults and contained some species not commonly identified in the adults oral cavity [55], suggesting that the oral microbiome dynamically changes and develops as an individual ages. Therefore, future studies should consider these multiple variations that may affect oral microbiome outcomes.

The mechanism by which the disease changes the oral microbial composition is still unknown; however, oral homeostasis has gained attention in neurological disorders [4]. The oral cavity is maintained at a relatively stable temperature of 35–37 °C, pH 6.5–7, and saliva production, which together provide an ideal environment for the growth of commensal oral microorganisms as well as a medium that delivers nutrients to bacteria [1]. Interestingly, neurological disease such as AD reduced saliva production and basal saliva pH, affecting the patient’s oral homeostasis [56,57]. Rozas et al. also found that PD patients with dysphagia, drooling, and lower salivary pH than healthy controls had significantly altered beta diversity and microbial composition, suggesting an important role of oral environment on the oral microbiome [11]. The alterations in oral ecology status, especially periodontal conditions, increased opportunistic pathogens such as *Spirochetes* (which includes *Treponema)* and *Prevotella nigrescens* over the commensal bacteria [58]. *Treponema* species can utilize amino acids (e.g., serine, alanine, cysteine, and glycine) and generate the fermentation products such as acetate, lactate, and pyruvate [58,59]. These compounds can influence the composition of bacterial species within the biofilms as well as affecting host tissue by penetrating the epithelial layers of the oral cavity [60]. *Prevotella nigrescens* was associated with Th17-mediated mucosa immune responses in vitro, by producing proinflammatory cytokines [42]. Findings from this current study suggest an alteration of oral microbiota composition and microbial diversity, and the assessment of changes in the oral microbiome may be an inexpensive and non-invasive biomarker for measuring progress of recovery in the patients with neurological diseases. Further studies are needed to elucidate detailed mechanisms of how changes in oral homeostasis are associated with disease pathogenesis.

The bidirectional communication between the oral microbiome and brain have not yet been defined. However, it has been proposed that the oral microbiota may affect the brain through direct or indirect means. Interestingly, *Treponema* species, the common oral bacteria, were detected in post-mortem AD brains [61], indicating a direct interaction of oral pathogens with the brain. AD patients also had increased levels of oral *Moraxella*, *Leptotrichia*, and *Sphaerochaeta* in saliva samples compared to healthy controls [32]. These Gram-negative bacteria release lipopolysaccharides (LPS), a strong stimulator of the immune response and inflammation, which are closely associated with worsening AD outcomes [62]. The levels of LPS and K99 pili protein released from Gram-negative bacteria were greater in the brains of AD patients than in normal brains [63], and intraperitoneal injection of LPS induced amyloid-beta plaques formation in rodent brain by stimulating neuroinflammation [64]. Gram-negative bacteria such as *Escherichia coli*, *Klebsiella* spp., *Kluyvera* spp., *Serratia* spp., *Proteus* spp., and *Enterobacter* spp. were also isolated in the oral cavity of PD patients [65]. Moreover, increased regional inflammation in the oral cavity was found in PD patients, showing elevated cytokine levels of IL-1 and IL-1RA compared to healthy controls [10]. These results suggest that an increase in oral pathogenic bacteria and their proinflammatory molecules may enter the brain tissue through the bloodstream, cause systemic and neuro-inflammation, and exacerbate disease severity and progression [44]. Therefore, oral dysbiosis may enhance neuropathology via the oral-brain connection. In addition, it is proposed that bidirectional communication between the TBI and oral microbiome may be mediated through immunological pathways. The secondary injury in TBI largely includes significant inflammation in the brain, such as the activation of microglia, the resident immune cell of the brain, and astrocytes. These activated cells in the brain produce a host of inflammatory cytokines (e.g., IL-1β, TNF-α) and chemokines (e.g., CCL2, CXCL10) that attracts peripheral immune cells to the brain, leading to an increasing cycle of inflammation [66,67]. The interconnectivity of the oral cavity and brain through the circulatory and lymphatic/glymphatic systems may potentially enable oral bacteria and inflammatory molecules to enter the brain, particularly since the blood–brain barrier is breached. Direct invasion of oral pathogens or inflammatory cytokines from the oral cavity into the brain may increase the inflammatory responses in the brain and exacerbate its severity or delay the recovery processes.

The top 10 most abundant genera in this study were *Streptococcus*, *Actinobacillus*, *Moraxella*, *Lactobacillus*, *Veillonella*, *Actinomyces*, *Rothia*, *Neisseria*, *Blautia*, and *Prevotella*, which is similar to a previous pig salivary microbiome study [30]. Oral and gut bacteria have different habitats; thus, they have distinct microbial diversity and structure in both humans [28,68,69] and pigs [30]. Maki et al. [28] summarized that the human oral microflora has a high abundance of *Streptococcus*, *Fusobacterium*, *Neisseria*, *Prevotella*, *Actinomyces*, *Pasteurella*, and *Veillonella*, whereas the gut has high abundance of *Alistipes*, *Akkermansia*, *Blautia*, *Faecalibacterium*, *Roseburia*, *Sutterella*, etc. [28]. The results from this study and others have shown that the pig oral microbiome was also enriched with *Streptococcus*, *Neisseria*, *Moraxella*, *Rothia*, *Actinobacillus*, and *Fusobacterium*, while earlier studies from our group and others have shown that fecal samples were abundant in *Clostridium*, *Lactobacillus*, *Turicibacteri*, and *Prevotella* [30,70]. This supports that the oral and gut microbiome have very different microbial composition. One similarity between the oral and gut microbiome is that different regions of the oral cavity, such as saliva, tonsils, buccal mucosa, gingiva, and plaque, have a distinct microbial composition [3], just as different gut segments have different microbial composition [71]. The existence of these various microbiomes suggests that it is important to understand how these diverse microbiomes of the human body interact with its host and affect health.

## 5. Conclusions

The findings of this study demonstrated that TBI induced changes in microbial diversity but did not alter the taxonomic composition of the oral microbiota during the acute stage of moderate/severe TBI in a swine model. The discovery of this phenomenon provides preliminary evidence for investigating the oral microbiome as a potential TBI biomarker and suggests future research directions. This is an early-stage study of the oral microbiota, and more research is needed to evaluate the dynamic interplay between TBI and the oral microbiome in longitudinal studies including the chronic TBI recovery stage.

## Figures and Tables

**Figure 1 brainsci-12-01111-f001:**
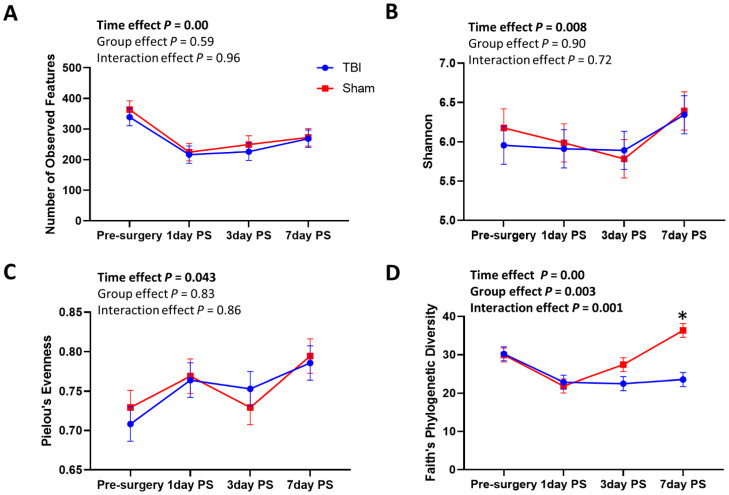
TBI did not significantly alter alpha diversity indexes of the oral microbiome except for Faith’s phylogenetic diversity. Alpha diversity indexes were measured by the number of observed features, Shannon, Pielou’s evenness, and Faith’s phylogenetic diversity. (**A**–**D**) Significant time effects were found in all alpha diversity indexes, while no group or time-by-group interaction effects were observed except for Faith’s phylogenetic diversity. (**D**) Faith’s phylogenetic diversity was significantly lower in the TBI group (*n* = 6) compared to the sham group (*n* = 6) at 7 days PS (time, group, and time-by-group interaction effects *p* < 0.05). *: Time-by-group interaction effect: Tukey post hoc comparison between the TBI and sham groups. PS, post-surgery.

**Figure 2 brainsci-12-01111-f002:**
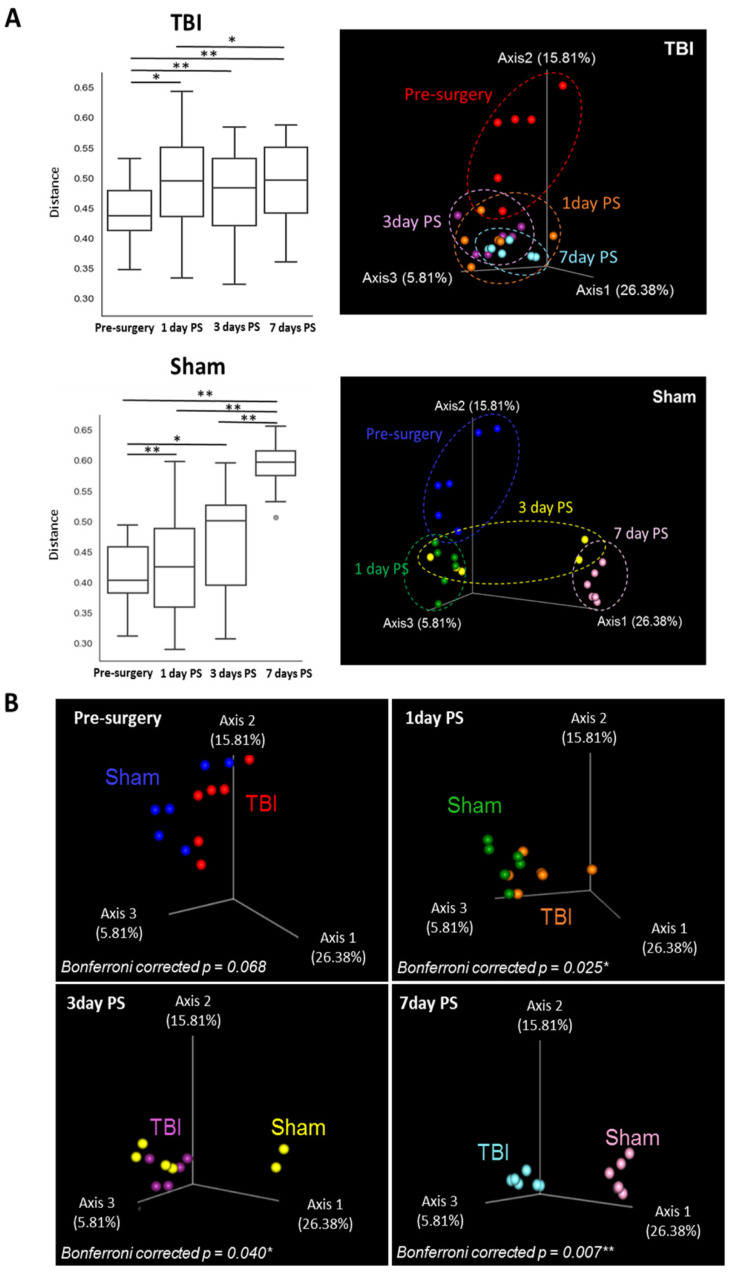
Beta diversity of the oral microbiome was different between TBI and sham piglets. Unweighted UniFrac matrix distance was used to evaluate the microbial pattern within and between groups during the acute stage of TBI. (**A**) Both TBI (*n* = 6) and sham (*n* = 6) groups showed significant changes in different beta diversity post-surgery compared to pre-surgery. (**B**) Distinct microbial patterns were observed between the TBI and sham groups at 1, 3, and 7 days PS, with the most apparent difference at 7 days PS. Bonferroni corrected *p*-value: * *p* < 0.05, ** *p* < 0.01. TBI, traumatic brain injury; PS, post-surgery.

**Figure 3 brainsci-12-01111-f003:**
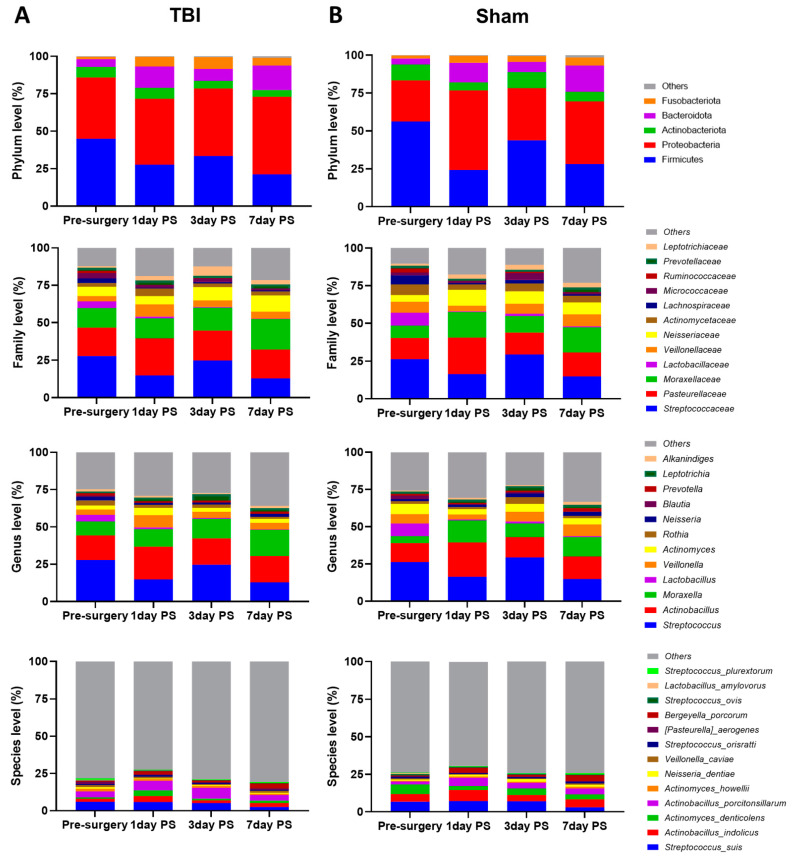
TBI did not significantly change the taxonomic composition of the oral microbiome between TBI and sham piglets. The oral microbial composition was analyzed at the phylum (>1%), family (>1%), genus (>1%), and species (>0.5%) levels pre-surgery and 1, 3, and 7 days post-surgery in (**A**) TBI (*n* = 6) and (**B**) sham (*n* = 6) groups. There were no significant compositional differences between TBI and sham groups up to 7 days post-surgery. TBI, traumatic brain injury; PS, post-surgery.

## Data Availability

All data in this study will be available from the corresponding author upon reasonable request.

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
