# Peer review of "Changes in Oral Microbial Diversity in a Piglet Model of Traumatic Brain Injury"

_brainsci, 2022, doi:10.3390/brainsci12081111_

Round 1

Reviewer 1 Report

Comments and Suggestions for Authors

The current paper evaluates the possible causative role TBI plays in the alteration of oral microbiome. The introduction provides a very brief background regarding the differences of oral and gut microbiome- this should be elaborated. Furthermore, the rationale for the study rests on observations made in patients suffering from chronic neurological diseases, however these studies were not designed to establish cause and effect relationship. The proposed mechanism by which an acute cerebral insult would cause change of oral microbiome should be elaborated. Additionally, the proposed significance of the hypothetical phenomenon should also be addressed.

The discussion touches this issue, by suggesting the possibility of trauma inducing inflammation, and the resultant inflammatory factors altering the oral microbiome, however no attempt was made to assess the level of these inflammatory mediators- the lack of this attempt should also be discussed.

The authors indicated that the small sample size is a limitation, which may in fact be the reason for the lack of significant change. It would be feasible to increase the sample size, so it will not be the reason for false negative results.

Author Response

The authors would like to thank the reviewers for their time and thoughtful consideration of the manuscript. We believe the insight provided by the reviewers has greatly increased the clarity of our manuscript and we appreciate the beneficial feedback. We addressed the reviewer’s comments point-by-point below and new additions and revisions to the manuscript are highlighted in yellow with track changes.

  1. The current paper evaluates the possible causative role TBI plays in the alteration of oral microbiome. The introduction provides a very brief background regarding the differences of oral and gut microbiome- this should be elaborated.

Response: The author would like to thank the reviewer for the comments. We described the differences between the oral and gut microbiome in the introduction and discussion of the revised manuscript.

  1. Furthermore, the rationale for the study rests on observations made in patients suffering from chronic neurological diseases, however these studies were not designed to establish cause and effect relationship.

Response: The authors agree with the reviewer’s comment that the findings from this study provide an association between the changes in oral microbiome and the condition of the diseases, instead of providing a piece of information on cause or effect relationship. We corrected the sentences in the Introduction accordingly (line 43, and 59-61).

  1. The proposed mechanism by which an acute cerebral insult would cause change of oral microbiome should be elaborated.

Response: Thank you for the valuable comment which enhances the quality and depth of the manuscript. The main focus of this study was to assess the changes in oral microbiota composition following experimental traumatic brain injury. The mechanism by which the TBI causes changes of oral microbiome is beyond the scope of this current project. However, the alteration of oral microorganisms has been observed in subjects with various diseases. We discussed more details with additional references in the Discussion. Newly added references are listed below. 

Larsen, J.M., The immune response to Prevotella bacteria in chronic inflammatory disease. Immunology, 2017, 151, p. 363-374.

Zalewska, A., et al., Salivary gland dysfunction and salivary redox imbalance in patients with Alzheimer’s disease. Scientific Reports, 2021, 11, p. 23904.

Aragón, F., et al., Oral health in Alzheimer’s disease: a multicenter case-control study. Clinical Oral Investigations, 2018, 22, p. 3061-3070.

Abusleme, L., et al., The subgingival microbiome in health and periodontitis and its relationship with community biomass and inflammation. The ISME journal, 2013, 7, p. 1016-25.

Dashper, S.G., et al., Virulence factors of the oral spirochete Treponema denticola. J Dent Res, 2011, 90, p. 691-703.

Kurita-Ochiai, T., K. Fukushima, and K. Ochiai, Volatile Fatty Acids, Metabolic By-products of Periodontopathic Bacteria, Inhibit Lymphocyte Proliferation and Cytokine Production. J Dent Res, 1995, 74, p. 1367-1373.

  1. Additionally, the proposed significance of the hypothetical phenomenon should also be addressed.

Response: Thank you for the suggestion and it has been added in the Conclusion of the revised manuscript.  

  1. The discussion touches this issue, by suggesting the possibility of trauma inducing inflammation, and the resultant inflammatory factors altering the oral microbiome, however no attempt was made to assess the level of these inflammatory mediators- the lack of this attempt should also be discussed.

Response: We agree with the comment. The inflammatory mediators such as circulating inflammatory cytokines will provide direct evidence supporting the hypothesis. As above mentioned, the focus of the current manuscript is on assessing the changes in oral microbiota composition in the swine model of TBI, and further mechanistic studies are warranted in a near future using this valuable animal model. We added this limitation in the discussion.

  1. The authors indicated that the small sample size is a limitation, which may in fact be the reason for the lack of significant change. It would be feasible to increase the sample size, so it will not be the reason for false negative results.

Response: Thank you for the critical comments. In the current preliminary study, the reason why we have a relatively small sample size is budgetary reasons. The study utilizing a large animal model with an injury that requires survival surgery is extremely expensive and intensive in budget and personnel. The findings of this still provide a new insight into the alterations in the oral microbiome in TBI which the authors find extreme value. We are currently seeking additional resources for a larger-scale study using this animal model with in-depth approaches to the mechanism.    

Reviewer 2 Report

Comments and Suggestions for Authors

The title of this article is “Changes in Oral Microbial Diversity in a Piglet Model of Traumatic Brain Injury”. This is an interesting topic, and needs our attention. However, there are still some areas of the article that need to be revised.

1. In the "Materials and methods" section of the article, the author needs to organize the content of this section appropriately, to make the article look more concise.

2. Figure 2. Beta-diversity of the oral microbiome was different between TBI and Sham piglets. In this part, the author may add more in-depth discussion and compare their results with the manuscripts recently published in authoritative journals.

3. In the "Discussion" section of the article, the authors mention that TBI causes changes in the composition of oral microorganisms, which is an interesting part to discuss in more depth according to the changed strains, pointing out the impact of the changes in oral microorganisms on health and TBI, and identifying the links between them.

4. The article points out that there is a link between oral microbiology and AD, and for this part, the authors need to think more deeply and give more of their own opinions and perspectives for the future.

Author Response

The title of this article is “Changes in Oral Microbial Diversity in a Piglet Model of Traumatic Brain Injury”. This is an interesting topic, and needs our attention. However, there are still some areas of the article that need to be revised.

Response: Thank you for reviewing the manuscript and providing your valuable comments. We believe that the comments increase the value and depth of the manuscript. We addressed the reviewer’s comments point-by-point and revised the manuscript.

  1. In the "Materials and methods" section of the article, the author needs to organize the content of this section appropriately, to make the article look more concise.

Response: We appreciate the reviewer’s suggestion. The Materials and methods section has been reorganized in the revised manuscript to make it more concise and clearer.

  1. Figure 2. Beta-diversity of the oral microbiome was different between TBI and Sham piglets. In this part, the author may add more in-depth discussion and compare their results with the manuscripts recently published in authoritative journals.

Response: Thank you for the valuable comments and the authors agree with the recommendation. We added a further description of beta-diversity in the Discussion with additional references as listed below.

Wagner, B.D., et al., On the Use of Diversity Measures in Longitudinal Sequencing Studies of Microbial Communities. 2018, 9, p.

Wingfield, B., et al., Variations in the oral microbiome are associated with depression in young adults. Scientific Reports, 2021, 11, p. 15009.

Simpson, C.A., et al., Oral microbiome composition, but not diversity, is associated with adolescent anxiety and depression symptoms. Physiology & behavior, 2020, 226, p. 113126.

Wagner, B.D., et al., On the Use of Diversity Measures in Longitudinal Sequencing Studies of Microbial Communities. 2018, 9.

Larsen, J.M., The immune response to Prevotella bacteria in chronic inflammatory disease. 2017, 151, p. 363-374.

Nagata, E. and T. Oho, Invasive Streptococcus mutans induces inflammatory cytokine production in human aortic endothelial cells via regulation of intracellular toll-like receptor 2 and nucleotide-binding oligomerization domain 2. Molecular Oral Microbiology, 2017, 32, p. 131-141.

Yang, I., et al., The oral microbiome and inflammation in mild cognitive impairment. Experimental Gerontology, 2021, 147, p. 111273.

  1. In the "Discussion" section of the article, the authors mention that TBI causes changes in the composition of oral microorganisms, which is an interesting part to discuss in more depth according to the changed strains, pointing out the impact of the changes in oral microorganisms on health and TBI, and identifying the links between them.

Response: We found in this study that there is a difference in microbial diversity in oral cavity between TBI and sham animals. An in-depth discussion of microbial diversity and diseases has been added to the Discussion of the revised manuscript. 

  1. The article points out that there is a link between oral microbiology and AD, and for this part, the authors need to think more deeply and give more of their own opinions and perspectives for the future.

Response: The approaches and understanding of oral microbiome changes and their impact on neurological disorders are novel and the studies are rare. Increased attention and accumulation of findings in this area are promising. We added further opinions and perspectives to the discussion.